# Uncertainty–guided learning with scaled prediction errors in the basal ganglia

**Moritz Möller**[1], **Sanjay Manohar**[1,2], **Rafal Bogacz**[1]*

**1** Nuffield Department of Clinical Neurosciences, University of Oxford, Oxford, United Kingdom,
**2** Department of Experimental Psychology, University of Oxford, Oxford, United Kingdom

* rafal.bogacz@ndcn.ox.ac.uk

## Abstract

To accurately predict rewards associated with states or actions, the variability of observations has to be taken into account. In particular, when the observations are noisy, the individual rewards should have less influence on tracking of average reward, and the estimate of the mean reward should be updated to a smaller extent after each observation. However, it is not known how the magnitude of the observation noise might be tracked and used to control prediction updates in the brain reward system. Here, we introduce a new model that uses simple, tractable learning rules that track the mean and standard deviation of reward, and leverages prediction errors scaled by uncertainty as the central feedback signal. We show that the new model has an advantage over conventional reinforcement learning models in a value tracking task, and approaches a theoretic limit of performance provided by the Kalman filter. Further, we propose a possible biological implementation of the model in the basal ganglia circuit. In the proposed network, dopaminergic neurons encode reward prediction errors scaled by standard deviation of rewards. We show that such scaling may arise if the striatal neurons learn the standard deviation of rewards and modulate the activity of dopaminergic neurons. The model is consistent with experimental findings concerning dopamine prediction error scaling relative to reward magnitude, and with many features of striatal plasticity. Our results span across the levels of implementation, algorithm, and computation, and might have important implications for understanding the dopaminergic prediction error signal and its relation to adaptive and effective learning.

## Author summary

The basal ganglia system is a collection of subcortical nuclei in the mammalian brain. This system and its dopaminergic inputs are associated with learning from rewards. Here, dopamine is thought to signal errors in reward prediction. The structure and function of the basal ganglia system are not fully understood yet—for example, the basal ganglia are split into two antagonistic pathways, but the reason for this split and the role of the two pathways are unknown. Further, it has been found that under some circumstances, rewards of different sizes lead to dopamine responses of similar size, which cannot be explained with the reward prediction error theory. Here, we propose a new model of

**Data Availability Statement:** All relevant data are within the manuscript and its Supporting Information files.

**Funding:** This work has been supported by Medical Research Council (MRC, mrc.ukri.org) grants

MC_UU_12024/5, MC_UU_00003/1 and
Biotechnology and Biological Sciences Research
Council (bbsrc.ukri.org) grant B/S006338/1 held by
RB, and an MRC clinician scientist fellowship MR/
P00878X to SGM. The funders had no role in study
design, data collection and analysis, decision to
publish, or preparation of the manuscript.

**Competing interests:** The authors have declared
that no competing interests exist.

learning in the basal ganglia—the scaled prediction error model. According to our model,
both reward average and reward uncertainty are tracked and represented in the two basal
ganglia pathways. The learned reward uncertainty is then used to scale dopaminergic
reward prediction errors, which effectively renders learning adaptive to reward noise. We
show that such learning is more robust than learning from unscaled prediction errors and
that it explains several physiological features of the basal ganglia system.

## Introduction

For any organism, better decisions result in better chances of survival. Reward prediction is an
important aspect of this—for example, if an organism can predict the size of a food reward
associated with some behavior, it can decide whether it is worth to engage in that behavior or
not. Reward predictions are typically based on values learned from previous reward observa-
tions. An extensive literature describes the role of reward prediction in behavior, as well as the
related neural mechanisms [1].

Piray and Daw [2] argue that when trying to predict rewards, the organism faces two chal-
lenges. The first challenge is the dynamic nature of the environment: reward sizes and contin-
gencies might change over time, in ways that cannot be predicted. Such genuine changes in
the environment can be quantified by the typical rate of change, which is called ***process noise***.
The second challenge is ***observation noise***: even if the environment is stable, rewards will vary
from experience to experience. This could be due to the random nature of the environment,
but also to variability in the organism's own behavior, or to noise in the organism's perception
and evaluation systems.

The stock market serves as a nice example of the two types of noise: consider the day–to–
day change of a stock price as a reward signal (if the stock price rises from 20 GBP to 21 GBP
overnight, then the shareholders win 1 GBP in that transition). Most of the variability of that
signal will be due to random fluctuations—this can be classified as observation noise. How-
ever, a part of the signal's variability will reflect genuine lasting changes in the stock prize, for
example a rise in price when a new product is released. This part should be classified as process
noise.

What is the best reward prediction method an organism could use when facing process
noise and observation noise? Similar problems occur in engineering, for example in the con-
text of navigation. There, a very versatile solution has been found. That solution, called the
***Kalman filter*** [3], is very widely used—it even played a role in the moon landing [4]. The Kal-
man filter describes how estimates of a variable must be updated when new noisy observations
of that variable become available. For certain types of signals, it can be shown that the Kalman
filter is indeed the ***optimal*** method for prediction in the presence of noise. The method has
proven useful not only in engineering, but also as a model of neural and behavioral processes
[5–8].

However, if one wants to use a Kalman filter to predict rewards, one runs into a problem:
the Kalman filter requires estimates of the magnitudes of both process noise and observation
noise as parameters. Where to take these values from? An organism might either use fixed
(perhaps genetically determined) values or estimate the values somehow. The former option
bears a risk: if the world changes, the quality of the organism's predictions might decline
strongly. The latter option raises the next question: how is this estimation done?

Solutions for this have been proposed. For example, Piray and Daw [9] present a model that
tracks both process noise and observation noise alongside reward, allowing for ***adaptive***

Kalman filtering. However, their model (a variant of the particle filter) is targeted at the computational level, i.e., it is set up to investigate how the simultaneous adaptation to two noise types of noise affects learning. Questions concerning the underlying biological mechanisms remain largely unaddressed. Hence it is unclear how the computations in the model of Piray and Daw [9] could be implemented in biological neural circuits.

This leads us to the central question of this paper: ***how might organisms track observation noise in a biologically plausible, computationally simple way, and use it for adaptive reward prediction?*** We propose that observation noise is tracked in the basal ganglia, and that it is used to improve learning performance by normalizing reward prediction errors. This proposal is based on the observation of dopamine activity patterns consistent with normalized prediction errors [10], as well as on previous suggestions that reward uncertainty might be represented in the basal ganglia [11,12]. The main novel contribution of this paper is proposing how the neural circuits in the basal ganglia can generate dopaminergic responses encoding learning signals, which are appropriately adjusted according to reward observation noise.

In the Results section, we give a detailed analysis of how the basal ganglia circuit might carry out the computations necessary to track and utilize reward observation noise. To provide some context for this analysis, we now move on to a brief review of the main features of that part of the brain.

## Basal ganglia system

Fig 1 shows a highly simplified version of the cortico–basal–ganglia–thalamic circuit, with three important brain regions—the cortex, the striatum, and the thalamus—arranged along the vertical axis.

The striatum is the largest nucleus within the basal ganglia system. It includes two populations of medium spiny projection neurons: the D1 and the D2 population (D1 and D2 are the types of dopamine receptors that the corresponding neurons express). This division of the striatum gives rise to two parallel descending pathways called the direct/Go and the indirect/No–go pathway, shown in green and red respectively in Fig 1 [13–15]. The cortical inputs to these two pathways are modulated by the strengths of the synapses between cortex and the striatal populations, collectively labeled $G$ and $N$ in Fig 1.

The thalamus receives the output of the basal ganglia circuit. The effects of the direct and the indirect pathway on the thalamus are differential: the direct pathway effectively excites the thalamus; the indirect pathway effectively inhibits it. Note that the projections from the striatum to the thalamus in Fig 1 are abstractions—in the brain, there are several intermediate nuclei between the striatum and the thalamus.

The final key element of the basal ganglia system are the dopamine projections from midbrain regions that target the striatal D1 and D2 populations. The effects of dopamine release on the striatal populations are twofold: dopamine modulates activity, but also triggers plasticity. The direction of those effects depends on the receptor type of the target neuron: dopamine increases excitability and potentiates synapses in the D1 population while it decreases excitability and depresses synapses in the D2 population [13–15].

Overall, we may view the basal ganglia circuit as two parallel descending pathways that converge on the level of the thalamus, where they have opposite effects. Those pathways are differentially modulated by dopamine, which also controls synaptic plasticity between the cortex and the striatum.

Concerning the function of the elements of this model of the basal ganglia, we follow a popular view often used in modelling [11,16,17]: the cortex supplies contextual information, i.e., cues, stimuli, sensory data or information on the state of the environment; the other

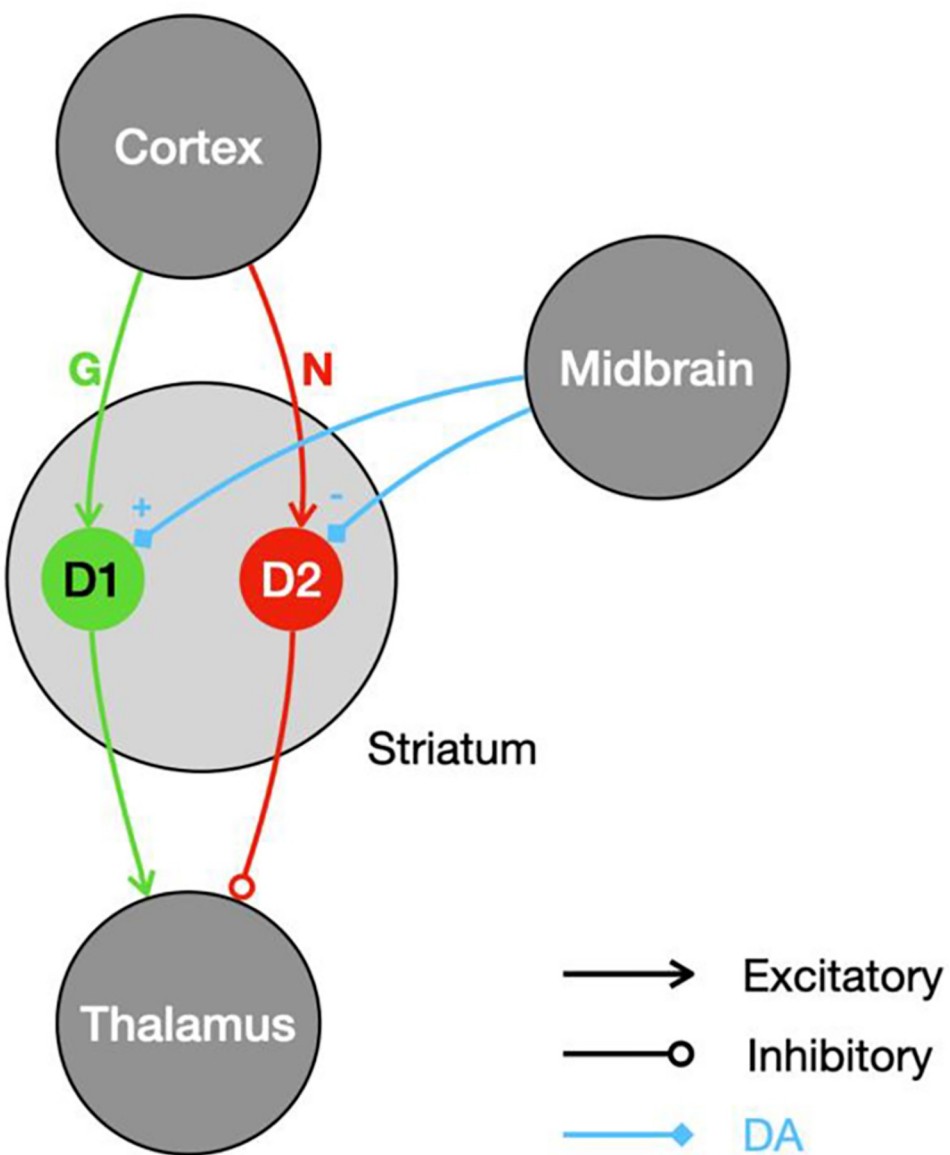

**Fig 1. The simplified basal ganglia circuit.** Selected nuclei and connections are shown as circles and arrows. Green connections correspond to the direct pathway; red connections correspond to the indirect pathway. Dopamine projections are shown in blue.

populations (D1, D2 and Thalamus) encode actions. Each action is represented by a distinct subpopulation of each nucleus, and the connectivity between the nuclei is action specific. For example, assume there is a subpopulation in D1 associated with pressing a lever. A corresponding subpopulation could be found in D2 as well as in the thalamus, which is known to relay motor commands to the relevant cortical areas [18]. The two striatal subpopulations associated with the lever press would then project exclusively to the lever–press subpopulation in the thalamus, together forming what is often called an action channel [19]. Learning is assumed to take place at the interface between the cortex and the striatum (which, in this model, can be considered a state–action mapping). Learning is implemented through dopamine–mediated plasticity of cortico–striatal synapses. These synapses within an action channel

are assumed to store information on the action value (the mean reward associated with the action). Action values determine the relative activations of action channels (i.e., the difference in activation between the Go–and the No–go pathways), and hence contribute to action selection at the level of the thalamus. It has been proposed [11] that cortico–striatal synapses additionally encode reward uncertainty (in the sum of the weights in the Go–and the No–go pathways), as we explain in detail below.

## Results

In the following sections, we present and analyze a model—the scaled prediction error (SPE) model—that tracks observation noise and uses it for adaptive reward prediction. In the first part of the Results, we introduce the model and test its performance. There, we show that it outperforms the classic Rescorla–Wagner (RW) model of associative learning [20] which does not adapt to observation noise, using simulations of a reward prediction task. In the second part, we discuss neural mechanisms that might implement the SPE model in the basal ganglia circuit. We first focus on dopamine signals, and then move on to the mechanisms behind tracking observation noise and scaling prediction error.

### The model

The SPE model is a model of reward prediction—it predicts the magnitude of the next reward based on previous reward observations. It can be understood as approximate Bayesian inference with respect to a particular model of the reward generation process. This model is given by

$$r_t \sim N(\mu, \sigma) \tag{1}$$

with $r_t$ the reward in trial $t$, $\mu$ the mean reward and $\sigma$ the reward observation noise. The SPE model can be derived by approximating Bayesian inference of the parameters $\mu$ and $\sigma$ from the observed rewards (we show this derivation in S1 Appendix). It does this by maintaining estimates $m$ and $s$ of those parameters, which it updates whenever a new reward is observed. Though the model is designed to infer the mean and standard deviation of stationary reward processes, it can also be applied to reward processes with a drifting mean (i.e., to processes with non–zero process noise). We show this below in our simulations.

The SPE update rules are

$$\delta = \frac{r_t - m_{t-1}}{s_{t-1}} \tag{2}$$

$$m_t = m_{t-1} + \alpha_m \delta \tag{3}$$

$$s_t = s_{t-1} + \alpha_s(\delta^2 - 1). \tag{4}$$

In these equations $\delta$ is a scaled reward prediction error, while $\alpha_m$ and $\alpha_s$ denote the learning rates for the mean and standard deviation, respectively.

At the start of the Method section, we show that the equations can indeed recover mean reward and its standard deviation, and here we summarize an intuition behind it. If $m$ and $s$ correctly estimate the mean and standard deviation of $r_t$, then the scaled prediction error of Eq 2 should have mean 0 and standard deviation 1. If the reward is higher (lower) than the estimated mean, then $\delta$ is positive (negative), and the estimated mean is increased (decreased) according to Eq 3. If the reward is further (closer) from $m$ than the estimated standard deviation, then $|\delta|$ is higher (lower) than 1, and the estimated standard deviation is increased

(decreased) according to Eq 4. Thus with time, $m$ and $s$ converge to the vicinity of mean and standard deviation respectively.

We further show in the Methods that the SPE model is related to the Kalman filter—it can be viewed as an approximation to a steady–state Kalman filter, which becomes more accurate if observation noise dominates process noise. Also, note that the RW model is a special case of the SPE model (i.e., if $\alpha_s = 0$).

## Performance

How do the SPE rules compare to established rules such as the Rescorla–Wagner model with respect to accurate reward predictions? To compare the performances of SPE learning and RW learning, we apply both to a reward prediction task: sequences of rewards are generated according to $r_t \sim N(\mu_t, \sigma)$; additionally the mean reward evolves according to $\mu_{t+1} \sim N(\mu_t, v)$, where $v$ is the standard deviation of the process noise. Both learners observe the reward signal and provide reward predictions at every trial. The learners' performance is judged by measuring the average precision of their predictions.

The task is designed to challenge the learners with rewards that change over time, forcing them to continuously learn. Note that the reward–generating process here is more complex than the generative model from which SPE learning is derived (Eq 1). This is not a problem— the SPE model is robust with respect to violation of its assumptions, as we shall see below.

It is important to stress that it is not our goal to show that the SPE model is optimal in this task (this would be surprising, since its derivation is not tailored to the reward generating process). Instead, we are interested in the SPE learning rules as a model of basal ganglia learning, and simply want to test their performance. Of course, there are learning rules tailored to this exact reward process: the Kalman filter rules (see Eq 17–Eq 19 in Methods). These rules involve an explicit representation of the process noise, or volatility. We will use this model as a normative benchmark, i.e., an upper bound on performance.

We compare the models for different levels of observation noise $\sigma$, while keeping the process noise $v$ constant at $v = 1$. The results of those comparisons are presented in Fig 2 (see Methods for details of the implementation).

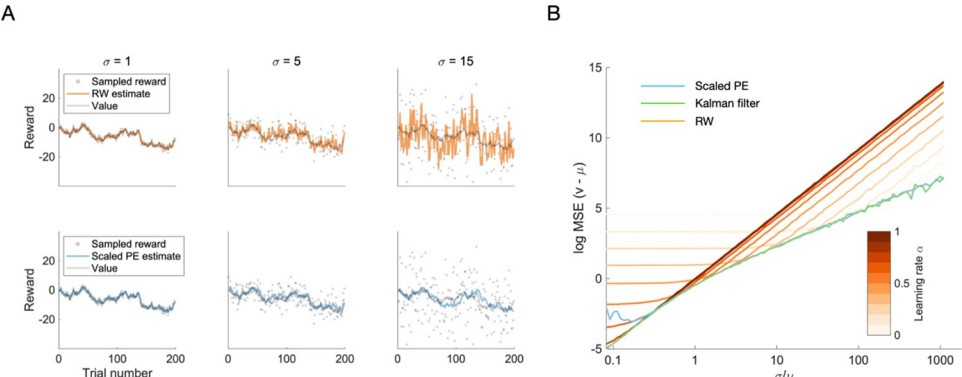

**Fig 2. Reward prediction performance of the RW, SPE and Kalman filter models. A** The first 200 trials of reward prediction for the RW learner (upper row, orange color) and the SPE learner (lower row, blue color). The true value (grey line), the observed rewards (grey dots) and the learner's estimate (colored line) are shown as a function of trial number. Columns correspond to selected levels of observation noise ($\sigma = 1, 5, 15$). **B** Learning performance averaged over trials. We show the logarithm of the mean squared difference between the mean of the reward distribution and the learner's prediction thereof, as a function of the observation noise $\sigma$. Orange lines correspond to RW learners, the blue line corresponds to a SPE learner parametrized with $\alpha_m = 1$ and $\alpha_s = 0.01$, and the green line corresponds to a Kalman filter parametrized with the true underlying process and observation noise parameters. The different shades of orange correspond to different learning rates, as indicated by the color bar.

Looking first at the time series in Fig 2A, we find that there is a qualitative difference between the RW learner in the top row and the SPE learner in the bottom row: as the noise level $\sigma$ increases, the RW learner's predictions increasingly fluctuate, since the reward prediction errors (and hence the updates) scale proportionally to the amplitude of the observation noise. This is not so for the SPE learner, whose predictions fluctuate as much for low noise levels as they do for high noise levels.

This effect is also visible in the aggregated performance measure, shown in Fig 2B: the mean squared errors of the learners' predictions grow with observation noise for all learners, but they grow stronger for the RW learners. We find a very stereotyped effect for the average performance of RW learners: as the level of noise increases, prediction accuracy does not change much up to a certain point (call this the plateau) and grows steadily after that point (call this the slope). This is the case irrespective of the learning rate. Smaller learning rates have a plateau that extends to higher noise levels but also provides a lower accuracy.

To gain an intuition for the shape of these curves, let us compare two different situations. First, consider very low levels of observation noise. In this regime, reward observations are almost identical to observation the underlying mean reward. Hence, if the observed reward changes, this mostly reflects a genuine change of the underlying mean reward. To keep reward predictions precise, such changes should be followed. However, for learning rates smaller than one, the RW model does not fully follow the changes of the reward signal—we may call this underfitting, as the model ignores meaningful variation in the signal. The resulting error dominates the performance. The magnitude of this error depends on the volatility of the signal. Since the volatility is kept constant in the simulations in Fig 2B, we see a performance plateau at low levels of observation noise.

Now, consider very high levels of observation noise. In this case, reward observations are very inaccurate—an observation tells us very little about the underlying mean reward. Changes in observed rewards mostly reflect the noisiness of the observations and should be ignored. However, for learning rates larger than zero, the RW model does not fully ignore those fluctuations, but follows them. This can be called overfitting, as the model tries to adapt to random fluctuations.

Overall, the behavior of the RW learners in Fig 2B is such that for each given level of observation noise there is an optimal learning rate, and the higher the observation noise, the lower the optimal learning rate. This appears consistent with intuition—if observation noise is high, there is less useful information in any single observation and an organism should therefore update its estimate more carefully.

On the other hand, Fig 2B shows that the SPE learner achieves performance about as good as the of the best RW model, for any given level of noise $\sigma$ larger than one. This suggests that in the regime of high observation noise we might view the SPE model as an RW learner that reaches optimal performance by fine–tuning itself to the estimated level of observation noise.

Can one do better than this? In fact, one can show that the SPE model (parametrized with $\alpha_m = 1$) is approximately optimal in the situation investigated here: for high levels of observation noise, SPE learning approximates the steady–state Kalman filter (see Methods), which is approximately optimal for the types of signals we use here. We also show the Kalman filter performance in Fig 2B (green curve). We find that in the regime of high observation noise, the performance of the Kalman filter (which is optimal for this type of signal) and that of the SPE model are very similar. Differences appear in the high process noise regime, where the SPE learner's performance deteriorates relative to the optimum set by the Kalman filter.

However, note that to use a Kalman filter, one needs to provide it with the correct values of $\sigma$ and $v$. This is also true for the steady–state version of the Kalman filter, but it is not the case for the SPE model: here one only needs to provide $\alpha_m$—which corresponds to $v$ (see Methods),

—but not $\sigma$, which the model can track by itself. We can thus think of SPE learning as ***adaptive*** steady–state Kalman filtering.

In summary, we find that the SPE model is approximately optimal for signals with $v<\sigma$. In particular, it will be at least as good as any RW learner, and about as good as a Kalman filter, even though the Kalman filter depends on a priori provision of $\sigma$, while SPE can infer it. SPE learners thus appear particularly well suited to track signals with high but unknown or changing levels of observation noise, as they can adapt themselves to whatever level of noise they experience. In contrast, an RW learner would either have to be fine–tuned based on prior knowledge, or it would perform suboptimal due to under–or overfitting. A Kalman filter would perform similarly to SPE, but only if its parameters are set correctly.

### The neural implementation of SPE learning

Could the SPE learning rules be implemented in the dopamine system and the basal ganglia pathways? In this section, we propose a possible mechanism. We suggest that striatal dopamine release broadcasts scaled prediction errors, $\delta = \frac{r-m}{s}$, and that the update rules given in equations Eq 3 and Eq 4 are implemented by dopamine–dependent plasticity in the striatum. In the next subsections, we will analyze the plausibility of these suggestions. First, we discuss the relationship between dopamine responses and scaled prediction errors. Then we discuss how the SPE learning rules can be mapped on striatal plasticity rules. Finally, we propose a mechanism that might implement the scaling.

### Scaled prediction errors are consistent with dopamine activity

In a seminal study, Tobler, Fiorillo [10] investigated how the responses of dopamine neurons to unpredictable rewards depended on reward magnitude, using electrophysiology in monkeys. Three different visual stimuli were paired with three different reward magnitudes (0.05 ml, 0.15 ml and 0.5 ml of juice). After being shown one of the stimuli, the monkeys received the corresponding reward with a probability of 50%. Seeing the stimulus allowed the monkey to predict the magnitude of the reward that could occur, but not whether it would occur on a given trial. Reward delivery thus came as a surprise and evoked a dopamine response. Interestingly, these responses did not scale with the magnitude of the received rewards. The measured dopamine responses are shown in Fig 3B.

This result was unexpected—standard RW learning would predict that the residual prediction errors in rewarded trials should grow linearly with reward magnitude. Our new SPE rules, on the other hand, predict exactly what has been observed. See Fig 3B for simulated and experimental DA responses.

One may object that the results of Tobler, Fiorillo [10] might also be explained by scaling with respect to the reward range—reward range and reward standard deviation cannot be dissociated in that experiment. While that is true, another recent experiment can dissociate them: Rothenhoefer, Hong [21] used two reward distributions with the same reward range but different reward standard deviations in a Pavlovian conditioning task (see Fig 3C).

After exhaustive training, single unit recordings were performed to measure dopamine responses to rewards that deviated from the expected value. It was found that the same deviation from the expected value caused stronger dopamine responses for the distribution with the smaller standard deviation (Fig 3D, first panel). This is consistent with scaling by reward standard deviation, but not with scaling by reward range–––both distributions had the same range, so scaling by range should yield similar responses for both conditions. These experimental data cannot be accounted for by the RW model (Fig 3D, second panel), but can be reproduced by the SPE model (Fig 3D, third panel).

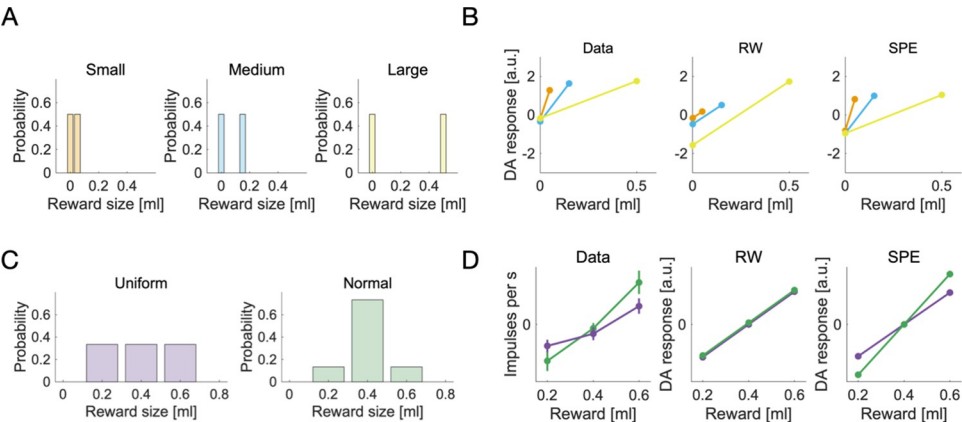

**Fig 3. Dopamine responses to unpredictable rewards—experimental data and simulations. A** The reward distributions used by Tobler, Fiorillo [10]. Each distribution corresponds to an experimental condition. **B** Dopamine responses to rewards sampled from the distributions in A are shown as a function of reward magnitude, for the three different conditions. The representation of data is similar to that in figure 4C of Tobler, Fiorillo [10]. We show experimental data, extracted from figure 4C (animal A) of Tobler, Fiorillo [10] and simulated data, using a standard RW model and the SPE model. The colors relate the dopamine responses in B to the reward distributions in A. **C** The reward distributions used by Rothenhoefer, Hong [21]. The panel shows the probabilities plotted by Rothenhoefer, Hong [21] in figure 1A. **D** Dopamine responses to rewards sampled from the distributions in C. We show the empirical values plotted by Rothenhoefer, Hong [21] in figure 2E, and the responses according to the RW model computed analytically as $\delta = r - \mu$, and the SPE model computed as $\delta = \frac{r-\mu}{\sigma}$, where $\mu$ and $\sigma$ are the mean and standard deviation of corresponding reward distributions in C. Purple lines correspond to the uniform reward distribution, green lines correspond to the normal reward distribution.

## Implementation of SPE learning rules in the basal ganglia

After establishing that dopaminergic scaled prediction errors are plausible, we now move on to discuss how the update rules given in Eq 2 and Eq 4 could be implemented in the basal ganglia circuit.

Mikhael and Bogacz [11] proposed a distributed encoding of the two reward statistics ($m$ and $s$) in the two main basal ganglia pathways: in their model, the mean of the reward signal is encoded in the difference between synaptic inputs to striatal neurons in direct (Go) and indirect (NoGo) pathways, whereas the standard deviation of the signal is encoded in the sum of these inputs. Mikhael and Bogacz [11] demonstrated that the mean and the spread of rewards can be learned with simple rules of striatal plasticity, according to which a positive prediction error mainly increases the weights of Go neurons, while a negative prediction error mainly increases the weights of NoGo neurons. Then, a high reward variability will lead to both positive and negative prediction errors and thus lead to increase of both sets of weights, so the reward variability will be reflected in the sum of Go and NoGo weights. Below we show that an analogous plasticity rules can also estimate mean and spread of rewards if the prediction errors are scaled by uncertainty, but these rules need to be slightly adjusted to be able to "read out" the scaled prediction error.

We assume the following relationship between the statistics of reward distribution and the weights in the direct and indirect pathways:

$$m = \frac{1}{2}(G - N) \tag{5}$$

$$\lambda(s - 1) = \frac{1}{2}(G + N) \tag{6}$$

In Eqs 5 and 6, $G$ and $N$ denote the synaptic inputs in the direct and indirect pathway respectively [22], and $\lambda$ is a coefficient determining the accuracy with which the standard deviation can be encoded (as explained below). Term –1 in Eq 6 in necessary for the neural implementation of scaled prediction error computation (as will be seen later). These assumptions can be used to rewrite the learning rules given in Eqs 3 and 4 in terms of $G$ and $N$. In particular, note that by combining Eqs 5 and 6, we see that $G = m+\lambda(s-1)$ and $N = \lambda(s-1)-m$. Therefore, we can derive the update rules for $G$ and $N$, namely $\Delta G = \Delta m+\lambda\Delta s$ and $\Delta N = \lambda\Delta s-\Delta m$. Hence by adding or subtracting Eqs 3 and 4, we obtain

$$\Delta G = \alpha_m f(\delta) - \lambda\alpha_s \qquad\qquad 7$$

$$\Delta N = \alpha_m f(-\delta) - \lambda\alpha_s \qquad\qquad 8$$

where $f$ is a non–linear transformation of prediction error (analogous to that in a previous model of striatal learning [11,23]):

$$f(\delta) = \delta + \lambda\frac{\alpha_s}{\alpha_m}\delta^2. \qquad\qquad 9$$

Using Eqs 5 and 6, the scaled prediction error can be expressed in terms of $G$ and $N$:

$$\delta = \frac{r - \frac{1}{2}(G - N)}{1 + \frac{1}{2\lambda}(G + N)} \qquad\qquad 10$$

It is worth emphasizing that Eqs 7–10 are equivalent to Eqs 2–4, because they are just rewritten in terms of different variables. Therefore, the model described by Eqs 7–10 estimates exactly the same mean and standard deviation as the model described by Eqs 2–4, and hence it produces identical performance in Fig 2 and dopaminergic responses in Fig 3.

One important issue while considering biological plausibility of the model is the fact that the synaptic weights on the indirect pathway $N$ cannot be negative, while the model assumes that these weights encode $N = \lambda(s-1)-m$. Imposing a constraint of $N$ being non–negative will limit the ability of the network to accurately estimate standard deviation of rewards to cases when it is sufficiently high (i.e. $\sigma\geq\mu/\lambda+1$). Hence the parameter $\lambda$ controls the accuracy with which the standard deviation can be estimated. However, according to Eq 6 there is a cost of high accuracy, because a high value of $\lambda$ will result in overall larger values of the synaptic weights (analogously as in the model of Mikhael and Bogacz [11]), and hence higher metabolic cost of the computations.

In the next two subsections, we analyze the biological plausibility of learning rules (Eqs 7–9), and computation of scaled prediction error (Eq 10).

## The SPE learning rules are consistent with striatal plasticity

Eqs 7 and 8 show three main features: 1) different overall effects of dopamine on plasticity in each pathway, 2) nonlinear effects of dopaminergic prediction errors represented by the transformations $f$ and 3) synaptic unlearning represented by decay terms. We will discuss the experimental data supporting the presence of these features in turn.

First, the efficacy of direct pathway synapses is assumed to increase as a result of positive reward prediction errors (i.e., $\delta>0$), and decrease as a result of negative reward prediction errors (i.e., $\delta<0$). The opposite is assumed to hold for indirect pathway synapses: their efficacy should decrease with positive prediction errors and increase with negative prediction errors. This premise corresponds to the sign of the prediction error in Eqs 7 and 8, and it is consistent with data obtained in experiments [24].

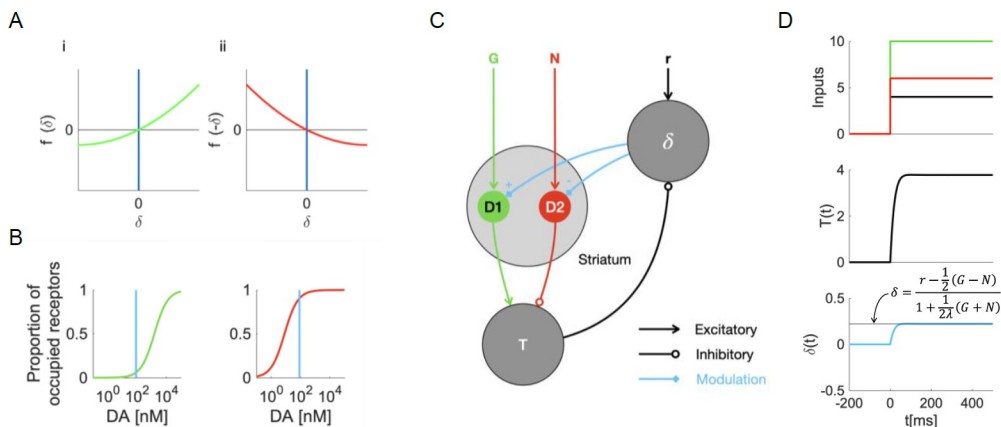

**Fig 4. Plasticity and computations in the basal ganglia circuit. A** The nonlinear transformation of dopaminergic prediction errors in the SPE model. The transformation in the direct pathway (i) and the transformation in the indirect pathway (ii) are mirror images of each other. **B** We plot the proportion of occupied receptors in the striatum as a function of dopamine concentration. The curves are based on the results of Dreyer, Herrik [26]. The blue vertical lines indicate the baseline dopamine concentration in the ventral striatum, based on the results of Dodson, Dreyer [27]. The green curve corresponds to the occupancy of D1 receptors, the red curve corresponds to the occupancy of D2 receptors. Panel B is adapted from figure 3D of Möller and Bogacz [23]. **C** The connectivity underlying a dynamical model of the simplified basal ganglia circuit. Circles correspond to neural populations; arrows between them indicate connections. **D** The computation of a scaled prediction error in continuous time, according to a dynamical model of the basal ganglia. We show how the relevant variables, $T$ and $\delta$, evolve as a function of time, assuming a step–function activation for the input nodes $G$, $N$ and $r$. The black line in the lowest panel indicates the level of dopamine required for exact SPE learning.

Second, it is assumed that for striatal neurons in the direct pathway, positive prediction errors have a stronger effect on plasticity than negative prediction errors. This assumption is expressed in the shape of the function $f(\delta)$, which is plotted in Fig 4Ai. Note that the slope for positive $\delta$ is steeper than for negative $\delta$, implying that positive prediction errors should lead to bigger changes in $G$ than negative prediction errors. The computational role of this nonlinearity is to filter the reward prediction errors: it amplifies the positive components while dampening the negative components of the signal. For striatal neurons in the indirect pathway, the SPE model assumes the opposite: negative prediction errors should have a stronger plasticity effect than positive prediction errors, because the weight modification is proportional to $f(-\delta)$, which is plotted in Fig 4Aii. Mikhael and Bogacz [11] argue that this premise is realistic, based on the different affinities of the D1 and D2 receptors that are present in striatal neurons in the direct and indirect pathways respectively: while D1 receptors are mostly unoccupied at baseline dopamine levels, D2 receptors are almost saturated—this is visualized in Fig 4B. Due to this baseline setting, additional dopamine should lead to a large difference in the occupation of D1 receptors and hence affect the neurons on the direct pathway, but only a small change in the occupancy of D2 receptors thus little influence the neurons on the indirect pathway. A decrease in dopamine, on the other hand, is strongly felt in D2 receptor occupancy but does not change D1 receptor occupancy much.

Third, an activity–dependent decay (or 'unlearning') is assumed to occur in the synaptic weights whenever they are activated in the absence of prediction errors. This is reflected in terms $-\lambda\alpha_s$ in Eqs 7 and 8. On the neural level, that premise translates into mild long–term depression after co–activation of the pre–and postsynaptic cells at baseline dopamine levels. Recently, this effect has been observed at cortico–striatal synapses in vivo [25]: in anaesthetized rats, presynaptic activity followed by postsynaptic activity caused LTD at baseline dopamine levels (i.e. in the absence of dopamine–evoking stimuli).

In summary, we discussed the three premises of the learning rules—the different overall effects of dopamine on plasticity in each pathway, the nonlinear effects of dopaminergic prediction errors and synaptic unlearning. We saw that all three premises are supported by the physiological properties of striatal neurons on the direct and indirect pathway.

## Computation of scaled prediction errors in the basal ganglia circuit

Next, we propose how the scaled prediction error in Eq 10 might be computed in the basal ganglia system. Eq 10 seems to be a complicated combination of terms, and it is difficult to see how a simple network might compute it. Surprisingly, this computation arises naturally in a feedback loop formed by the dopaminergic neurons and the striatum. In particular, the dopaminergic neurons compute the prediction error based on expected reward encoded in striatum, but their activity also modulates the gain of the striatal neurons (Fig 4C). We show below that this modulation in turn changes the signal sent by striatum to dopaminergic neurons and introduces scaling of prediction error. We will describe that mechanism, using a minimal dynamical model of the basal ganglia network.

A population of dopaminergic neurons encoding the prediction error $\delta$ needs to receive excitatory input $r$ that encodes a reward signal, and inhibitory input reflecting utility of expected reward. We assume such utility is encoded in thalamic activity, which we denote by $T$. Formally, we assume:

$$\delta = r - T \qquad 11$$

In the basal ganglia circuit the information on expected rewards could be provided to dopaminergic neurons by other projections, but as the anatomical details of these projections are still unclear, we conceptualize them as an input from thalamus for simplicity. We follow Möller and Bogacz [23] in assuming that the thalamic activity reflects the total output from the basal ganglia —the difference between the activity in the direct and indirect pathways—which is captured by

$$T = \frac{1 + \delta/\lambda}{2} G - \frac{1 - \delta/\lambda}{2} N \qquad 12$$

The first term $\frac{1+\delta/\lambda}{2} G$ corresponds to the activity in the direct pathway, which is proportional to synaptic input $G$, and is increased by the dopaminergic modulation, because the gain of striatal neurons in the direct pathway is enhanced by dopamine. The second term $\frac{1-\delta/\lambda}{2} N$ corresponds to the activity in the indirect pathway, which is attenuated by dopamine, because the gain of striatal neurons in the indirect pathway is reduced by dopamine [13,14]. The proposed model contains a feedback loop: dopamine release modulates the thalamic activity, which itself inhibits dopamine release.

To examine the computation of scaled prediction errors, we model the relevant populations' activities as leaky integrators with effective connectivity as sketched in Fig 4C, using differential equations in continuous time. The dynamical system sketched in Fig 4C corresponds to a set of differential equations,

$$\tau_\delta \dot{\delta} = -\delta + (r - T) \qquad 13$$

$$\tau_T \dot{T} = -T + \frac{1 + \delta/\lambda}{2} G - \frac{1 - \delta/\lambda}{2} N \qquad 14$$

Here, $\tau_\delta$ and $\tau_T$ are the characteristic timescales of the striatal dopamine release and thalamic activation. The system is set up such that its equilibrium point is consistent with our

trial–wise description, i.e. at $\dot{\delta} = \dot{T} = 0$, the prediction error and thalamic activity are given by Eqs 11 and 12. This asserts that the two levels of description are consistent with each other. Using these equilibrium equations, we can determine the equilibrium value of $\delta$ (by inserting Eq 12 into Eq 11 and solving for $\delta$). We find it is given by the scaled prediction error in Eq 10. This suggests that the circuit can compute the scaled prediction error. The computation will be accurate as long as $s$ is sufficiently large so it can be encoded in non–negative $G$ and $N$ according to Eq 6. For lower $s$, the term 1 in the denominator of Eq 10 ensures that the denominator is not too small and it prevents catastrophically large prediction errors that might cause the instabilities.

So far, it looks as though the circuit has an equilibrium point at approximately the right value. However, it is not yet clear whether and how this equilibrium is reached. To learn more about these aspects, we need to simulate the system. To simulate the computation of the prediction error, we assume $G$, $N$ and $r$ to be provided externally, for example through cortical inputs. $G$ and $N$ then represent precisely timed reward predictions, while $r$ represents the reward signal itself. We model $G$, $N$ and $r$ as step–functions that jump from zero to their respective values at the same time, as illustrated by the first panel of Fig 4D. The time constants $\tau_\delta$ and $\tau_T$ are set to realistic values taken from the literature (see Methods). A simulation of the system is shown in Fig 4D. We find that $\delta$ settles to its equilibrium value quite quickly (after tens of milliseconds) and without oscillations. This is likely due to the difference in time constants—the thalamic activity changes much faster than the striatal dopamine concentration. Our results suggests that even a simple system as the one in Fig 4C can compute scaled prediction errors through a feedback loop.

## Discussion

Above, we presented a new model of error–driven learning: the SPE model. We tested it in simulations and compared it with neural data. Now, we will discuss the new model more broadly. First, we will summarize our key findings. Then, we will present several testable predictions that follow from the model. Finally, we will discuss how the SPE model relates to other models.

### Summary

This work introduces the SPE model, which describes how an organism might adapt its learning mechanism to changing levels of reward observation noise $\sigma$. First, we proposed the SPE learning rules, which can track the mean and standard deviation of a reward signal. We then tested the performance of the new rules. Comparing SPE learning with RW learning, we found that the new learning rules can improve performance when a learner faces unknown or varying levels of reward observation noise. Next, we reviewed empirical evidence relating to SPE learning. On the neural level, we found that SPE learning describes dopamine responses better than conventional models in several studies. We further showed how the basal ganglia pathways might implement the learning rules of the SPE model, and how scaled prediction errors could be computed in a dopaminergic feedback loop.

### Experimental predictions

Our model makes several predictions on different levels of analysis. First, SPE learning can be distinguished from other types of learning on the level of behavior. This is because according to SPE, the learning rate (and hence the speed of learning) should depend on the stochasticity of the reinforcement that drives learning. From this, different predictions follow.

For example, reward magnitude should ***not*** affect instrumental learning speed if animals were exposed to the reward prior to the instrumental phase. This is because SPE learning

would allow the animal to normalize the reward magnitude through the adaptive scaling of prediction errors, as in the experiment by Tobler, Fiorillo [10]. The dopamine–guided instrumental learning through normalized prediction errors would then not be affected by the overall scale of the rewards. Concretely, this could be tested in a decision–making task with two conditions. In one condition, correct choices are reinforced with high rewards (for example 0.6 ml of juice). In the other condition, low rewards are provided, for example 0.2 ml of juice. Conditions must be cued, perhaps by two different visual stimuli that precede the decision. According to the RW model, we expect to see a steeper learning curve in the high reward condition, as in the experiment of [28]. However, SPE theory predicts that this difference between conditions should vanish if an appropriate pretraining is applied. For example, the cues could be associated with the different reward sizes through Pavlovian conditioning. According to SPE theory, the pretraining should establish a condition–specific normalization cued by the stimuli. This normalization should then lead to normalized learning curves in the instrumental phase. Ultimately, the difference between the learning curves should vanish.

Furthermore, SPE learning predicts that learning rates should change if reward stochasticity is changed. This could be tested by having participants track and predict a drifting reward signal which shows different levels of stochasticity at different times. If participants use SPE, the learning rate should decrease with increasing stochasticity, which would lead to invariant update magnitudes. On the other hand, if participants do not use SPE, increasing reward stochasticity would not affect the learning rate, and hence lead to larger updates.

On the neural level, SPE learning predicts trial–by–trial changes of how the dopaminergic prediction error is normalized to a given reward signal. Neural recordings from the relevant brain areas during the learning phase could be compared with simulations of the SPE model to test the theory. In particular, SPE predicts that if reward stochasticity increases slowly, the scale of the corresponding dopamine bursts should stay invariant. Standard theory, on the other hand, would predict that the scale of dopamine bursts grows proportional to the scale of rewards, as prediction errors are a linear function of rewards.

Since the SPE model is closely related to the AU model [11], it inherits a prediction on how the activity in the basal ganglia pathways should depend on reward stochasticity: if reward stochasticity is high, the sum of the activity in the direct and the indirect pathway should be high —if reward stochasticity is low, the sum should be low as well. It has been indeed observed that the neural activity in striatum increases with reward uncertainty [29,30]. Cell–type specific imaging techniques such as photometry could be used to further assert whether the uncertainty is encoded in the sum of activity of striatal neurons on the direct and indirect pathways.

## Relationships to other models

**Kalman filter.** We demonstrated that the SPE model is closely related to the Kalman filter and both models achieve similar performance. Although the Kalman filter has been originally developed for statistical and engineering applications, it has also been used to describe human reinforcement learning [31] and representation of uncertainty during decision making [32]. Therefore, it is useful to discuss how representation of uncertainty in the SPE model relates to that in the Kalman filter. The Kalman filter, in addition to representing observation and process noise, also represents a third type of uncertainty–the posterior uncertainty. It is the standard deviation of posterior distribution of mean reward given the observations, so it describes the agent's uncertainty about the current estimate of reward. The posterior uncertainty is considered by some researchers to be the most important facet of Kalman filtering, and it influences the optimal learning rate in the Kalman filter, hence it may seem surprising how the SPE model, which does not keep track of it, can achieve performance similar to the Kalman filter.

Below we explain why such performance can be achieved without explicitly representing posterior uncertainty.

The posterior uncertainty in the Kalman filter only changes at the beginning of learning and then it converges to a fixed value that depends on the process and observation noise (Fig 5A). Hence in the steady state, an agent could recover an estimate of the posterior uncertainty on the basis of its estimates of observation and process noise. Importantly, in the steady state, the optimal learning rate can be expressed as a function of observation and process noise alone (Fig 5B, blue), hence for optimal filtering beyond the initial trials, one does not need to explicitly represent the posterior uncertainty, but instead it is sufficient to encode the estimates of observation and process noise. The effective learning rate in the SPE model (Fig 5B, orange) is similar to that in the steady–state Kalman filter for higher levels of observation noise, which underlies the similar performance of the SPE model and the Kalman filter in Fig 2B.

It is also worth highlighting that the rewards uncertainty estimated by the SPE model, is not equal to the posterior variance. For example, if the process noise $v = 0$, the posterior variance converges to 0 (Fig 5A, green), while the estimate of observation noise in the SPE model will fluctuate around $\sigma$ (see Eqs 20 and 22 for the relationship between observation noise and the posterior variance).

The key difference between the Kalman filter and the SPE model is that the latter has a mechanism to track $\sigma$. No such mechanism exists in the Kalman filter. Both models require $v$ as an external input which corresponds to providing it with information about the level of volatility (for the SPE model, the corresponding parameter is $\alpha_m$). To make the model more autonomous, one might extend it with a mechanism to track $v$ alongside $\sigma$, for example the mechanism proposed by Piray and Daw [2]. This is an interesting direction for further research but goes beyond the scope of this work.

**Models that feature scaled prediction errors.** Our work here is not the first to address precision–weighted prediction errors. Previous research has investigated similar issues; several computational models have been proposed in this context. We now review some of the most important models and point out how our work differs from those accounts.

First, there is a series of studies that builds on the classical study of Tobler, Fiorillo [10], and systematically investigates scaled dopaminergic prediction errors [33–36]. This work shows

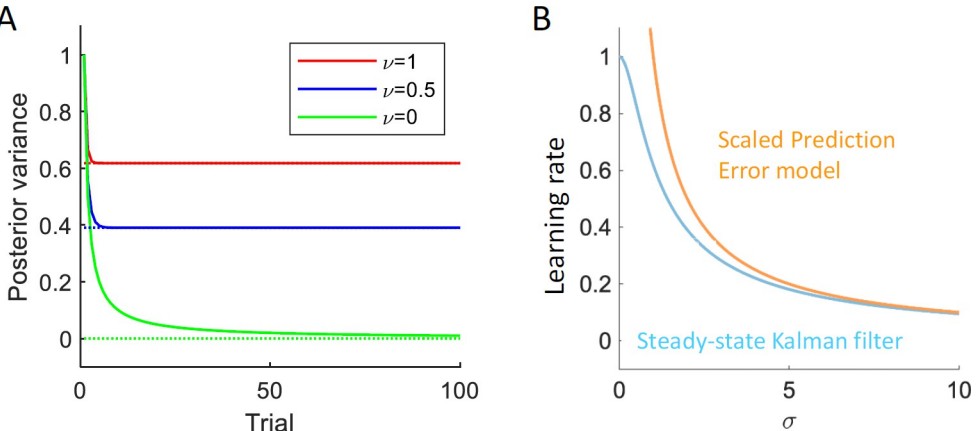

**Fig 5. Variables of the Kalman filter and its approximations. A** Posterior variance in the Kalman filter (solid, Eq 19) and the steady–state Kalman filter (dotted, Eq 20), as a function the number of observations. Different colours correspond to different levels of process noise, and the values are plotted for the standard deviation of the observation noise of $\sigma = 1$. **B** The learning rate of the steady–state Kalman filter $k_\infty$ (blue, Eq 21) and the approximation $k_\infty \approx \frac{v}{\sigma}$ (orange) which corresponds to the effective learning rate in the SPE model. We show the learning rates as a function of the observation noise $\sigma$ for process noise of $v = 1$.

that 1) theoretically, scaled prediction errors should lead to observable behavioral adaptation, which in turn improves performance in learning tasks; 2) that such behavioral adaptation can be observed in human behavior, and is indeed correlated with improved task performance; 3) that this empirical behavioral adaptation is indeed linked to scaled prediction error signals recorded from the dopaminergic midbrain using imaging techniques; 4) that this behavioral adaptation can be manipulated using DA targeting drugs and 5) that this mechanism is impaired in patients with psychosis. The evidence gathered in these studies thoroughly establishes the existence of dopaminergic scaled prediction errors, as well as a causal link of this scaling to behavioral adaptation and the corresponding increase in task performance.

While this research strongly supports the theory presented in this paper, there are clear differences: most importantly, the research in [33–36] does not investigate *how* the scaling is achieved, i.e., 1) how the scaling factor is learned in a biologically plausible way and 2) how the entire computation of scaled prediction errors is implemented in neural circuits. Another difference is the actual scaling factor, which is assumed to be the logarithm of the standard deviation rather than the standard deviation that is used in SPE. Due to these differences, it is important that we test our learning rules in simulations, and establish that they improve task performance even when the scaling is different and based on learned estimates of observation noise.

Another related study [37] focusses on behavior in a task with observation noise and discontinuous change points. There, a variety of models are derived to describe learning, and empirically tested. One of these models uses an estimate of observation noise which is learned with a delta rule. However, there is an important difference: the learned estimate of observation noise is not used to scale prediction errors, but instead enters the learning rate in a quite indirect way. Further, the entire study has its focus on sudden, non–continuous changes in the signal, and does not suggest concrete circuit implementations of the algorithmic models. We can thus view that study as complementary to our work here—the combination of the two models could be a very interesting avenue for future research.

One recent theory—Kalman–TD—explained the scaling of dopamine responses, as well as other phenomena such as preconditioning, as a consequence of volatility tracking [5]. Kalman–TD applies the Kalman filter method to the computational problem of TD learning: reward prediction in the time domain. The resulting model features vector–valued learning rates that constantly adapt to observations and outcomes. It elegantly describes how covariances between cues and cue–specific uncertainties might modulate learning and can be shown to explain several empirical phenomena. However, the Kalman–TD theory does not address the tracking of observation noise (the theory focuses on process noise). It also does not discuss how prediction error scaling might be implemented. We may thus view it as a complement rather than a competitor to the theory presented above.

**The reward taxis model.**   Another model was recently proposed to explain the effects reported by Tobler, Fiorillo [10] and other phenomena. The model is called ***reward taxis*** [38], and explains the dopaminergic range adaptation using a logarithm: if both rewards and reward expectations were transformed by a logarithmic function, prediction errors would be given by $\delta = \log r - \log m = \log \frac{r}{m}$. In the experiment of Tobler, Fiorillo [10] rewards were given in 50% of the trials. For a reward of size $r$, the expected reward would then be $m = \frac{r}{2}$, and the prediction error would be $\delta = \log \frac{r}{\frac{r}{2}} = \log 2$, i.e., independent of reward size. Reward taxis can hence explain the results of Tobler, Fiorillo [10] quite elegantly.

However, that explanation breaks down as we look at other experiments. We have already mentioned the experiment by Rothenhoefer, Hong [21], which featured two reward distributions with equal means and ranges but different standard deviations. We show those distributions in Fig 3C. Rothenhoefer, Hong [21] first used Pavlovian conditioning in a way similar

way to Tobler, Fiorillo [10], pairing the two reward distributions with two different cues. They then recorded the dopamine responses at reward delivery, for all reward sizes of each distribution. We reproduce their data in Fig 3D (first panel). The responses to the middle reward are similar for both distributions, but the responses to the extreme rewards differ: they seem scaled up for the normal distribution.

What would the reward taxis theory predict for the responses in this experiment? Both distributions have the same mean; reward taxis hence predicts similar responses for both distributions. The experimental data thus falsifies the reward taxis model in this experiment. In contrast, the SPE model predicts different responses for the two distributions—we show this in Fig 3D (last panel). Overall, it appears as if dopamine responses to reward distributions with variable width are better captured by the SPE model than the reward taxis model.

**Models learning reward uncertainty.** The SPE model is closely related to the AU model of Mikhael and Bogacz [11]—both models describe how the basal ganglia pathways track reward uncertainty; they also share the distributed encoding of reward statistics. It is thus not surprising that the learning rules of the two models have similarities. However, the SPE model differs from the AU model in several important aspects.

Of course, the scaled prediction error itself is the key new feature that drives most of the interesting effects we investigated in this work. It is through the scaling of the prediction error that our new model puts its estimate of the reward observation noise to good use. The AU model tracks reward noise as well but does not use its estimate to improve learning performance (or for anything else). In contrast, the SPE model explains not only ***how*** to track $\sigma$, but also ***why***.

Further, the AU model assumes that there are two separate dopamine signals that modulate activity and plasticity of striatal neurons, namely that the tonic level modulates activity, while the phasic bursts trigger plasticity. However, it has been recently demonstrated that even a brief, burst–like activation of dopaminergic neurons changes the activity levels of striatal neurons [39]. Additionally, it has been shown that reward prediction errors modulate the tendency to make risky choices [22], and risk attitudes are known to depend on the balance between the activities in direct and indirect pathways [40,41]. In this paper, we demonstrated that a more realistic assumption, that the dopamine signal encoding prediction error also changes the activity levels in striatum, enables scaling of prediction errors by uncertainty.

It has been also proposed that the basal ganglia system learns not only the mean and variance of the rewards, but the entire shape of reward distribution [12]. That model employs learning rules similar to those of the AU model in that the positive and negative prediction errors have different magnitudes of effects on plasticity in different neurons. While in the AU and SPE models there are just two types of neurons with different learning rules (Go and No–Go), in the distributional model [12] each neuron may have slightly different learning rule, resulting in different neurons learning different percentiles of the reward distribution. Although in some contexts it may be useful to know the shape of reward distribution, only its standard deviation is necessary to optimize effective learning rate to approximate Kalman filter. Therefore, it may be interesting to investigate how the information on standard deviation of reward could be extracted from that model and used to scale prediction error.

**Free energy models.** Finally, we want to discuss the relation of our model to free energy models: the scaled reward prediction errors in this work are formally related to the precision weighted prediction errors of the free–energy approach, especially when the recognition density (the learner's model of the world) is taken to be Gaussian [42–44]. In that case, the prediction errors that drive inference and learning in free energy models are often weighted by precisions, i.e., inverse variances. The connection to scaled reward prediction errors becomes very close when the free energy approach is applied to reward prediction, dopamine and the

basal ganglia system, as has been done in the DopAct model [45]. This model includes weighted prediction errors encoded by dopamine transients, but they are not the focus of the theory, and for simplicity they are modelled as in earlier free energy models [46]. In particular, it is assumed that the variance of observation noise is encoded in the rate of decay of dopaminergic neurons (or self–inhibitory connections among them), which results in dopaminergic neurons encoding prediction error divided by the variance of the observation noise. However, such mechanism would only be able to learn the overall variance of the observation noise averaged across all contexts and actions, because the dopaminergic neurons encode reward prediction error for all states. By contrast, the mechanism proposed here, where observation noise is learned in striatum, could potentially learn observation noise associated with individual sources of rewards, because striatal neurons are selective for different states and actions. Furthermore, it is important to note that precision, or inverse variance, scales differently to standard deviation, and might hence not explain classical observations such as those reported by Tobler, Fiorillo [10].

Free energy models (e.g. Mathys, Daunizeau [47]) have shown in various ways that tracking uncertainty is very relevant for successful learning, and have stressed the importance of precision weighting for performance. In this context, our performance results above are not surprising. However, it is important for us to show that our model (which is different and much less complex than the hierarchic gaussian filter, for example) still improves learning outcomes. Since our focus is on modelling learning in the basal ganglia and the dopamine system, our model should not be viewed as a competitor of this family of models, rather as a relative, applied to a specific circuit and type of signal. Nevertheless, scaling of prediction errors may be a fundamental and common mechanism in the brain, and it would also be an interesting direction for future work to investigate scaled prediction errors in other systems within the brain.

## Methods

This section first describes mathematical analyses of the SPE model, and then details of simulations in the paper.

### Fixed point analysis

Here, we show that in theory, $m$ and $s$ updated according to Eqs 2–4 should converge to the mean and the standard deviation of the reward signal. To demonstrate it, we use a stochastic fixed–point analysis. Let us assume that rewards are indeed generated by sampling from a distribution with mean $\mu$ and standard deviation $\sigma$ (this could be a normal distribution or any other distribution with well defined mean and standard deviation).

We consider a situation in which the learner has already found the correct values of the variables it maintains, i.e., $m = \mu$ and $\sigma = s$. From there, what are the **expected updates**? A straightforward calculation yields

$$E(\Delta m) = \alpha_m E\left(\frac{r - m}{s}\right) = \alpha_m \left(\frac{Er - \mu}{\sigma}\right) = \alpha_m \left(\frac{\mu - \mu}{\sigma}\right) = 0 \qquad 15$$

$$E(\Delta s) = \alpha_s \left(\frac{E(r - m)^2}{s^2} - 1\right) = \alpha_s \left(\frac{E(r - \mu)^2}{\sigma^2} - 1\right) = \alpha_s \left(\frac{\sigma^2}{\sigma^2} - 1\right) = 0 \qquad 16$$

with $Er = \mu$ and $E(r-\mu)^2 = \sigma^2$ by definition. We find that the expected change away from $(m, s) = (\mu, \sigma)$ is zero, which makes $(\mu, \sigma)$ a stochastic fixed–point. We may conclude that in

equilibrium, the rules given in Eqs 2–4 should give us unbiased estimates of the reward mean and standard deviation.

## The high noise limit of the steady state Kalman filter

Here, we show that the SPE learning rules approximate the one–dimensional steady–state Kalman filter in the limit of high observation noise. We start by defining the Kalman filter model. We then derive the steady–state Kalman filter, and finally take the high–noise limit.

## The definition of the Kalman filter

The Kalman filter is a computational method for state estimation and prediction [3]. It can be derived from Bayesian principles and is optimal for tracking signals with certain characteristics. Here, we focus on a one–dimensional Kalman filter which is used for predicting rewards, following Piray and Daw [2]. The rules they use are

$$m_t = m_{t-1} + k_t(r_t - m_{t-1})$$
(17)

$$k_t = \frac{w_{t-1} + v^2}{w_{t-1} + v^2 + \sigma^2}$$
(18)

$$w_t = (1 - k_t)(w_{t-1} + v^2)$$
(19)

where $r_t$ is the reward, $k_t$ the learning rate or Kalman gain and $w_t$ the posterior variance in trial in trial $t$. Note that our notation differs slightly from that of Piray and Daw [2], for the sake of consistency within this work.

The above rules can be shown to be optimal for tracking signals such as those we used above, i.e., signals that consist of samples drawn from a normal distribution with a drifting mean [3].

## The steady–state Kalman filter

The Kalman filter has several variables that must be updated on every trial. If one requires a simpler model with almost similar properties, one option is to use a Kalman filter in the limit $t \to \infty$: as for $t \to \infty$, the posterior variance $w_t$ and the Kalman gain $k_t$ converge to limits $w_\infty$ and $k_\infty$.

Eq 17 with $k_\infty$ instead of $k_t$ is called a ***steady–state Kalman filter***. By construction, the normal Kalman filter becomes more similar to the steady–state Kalman filter the more trials pass. In practice, performance often does not differ much between the two [3].

What are the limits $w_\infty$ and $k_\infty$? One may use Eqs 18 and 19 to determine them. By setting $k_t = k_{t-1}$ and $w_t = w_{t-1}$, we find

$$w_\infty = \frac{v^2}{2} \left( \sqrt{4\frac{\sigma^2}{v^2} + 1} - 1 \right)$$
(20)

$$k_\infty = \frac{\sqrt{4\frac{\sigma^2}{v^2} + 1} + 1}{\sqrt{4\frac{\sigma^2}{v^2} + 1} + 1 + 2\frac{\sigma^2}{v^2}}$$
(21)

To use the steady–state Kalman filter, one just needs to compute $k_\infty$ and plug it into Eq 17. One can then use this single equation to track the signal, with no other computations required.

The steady–state Kalman filter is thus equivalent to the RW model (Eqs 25 and 26), parametrized with an optimal learning rate (that is to say, optimal for a signal with statistics $v^2$ and $\sigma^2$).

## The high observation–noise limit

For the case of observation noise $\sigma$ higher than the process noise $v$, the expressions for $w_\infty$ and $k_\infty$ drastically simplify which provides insight on how they depend on $\sigma$ and $v$. Using Eq 20, we find that

$$w_\infty \rightarrow v\sigma \qquad\qquad 22$$

for $\frac{\sigma}{v} \rightarrow \infty$. In this limit, there exists a very simple relationship between the posterior variance in the steady state $w_\infty$, and the variances of observation $\sigma^2$ and process $v^2$ noise, i.e. the posterior variance is approximately equal to the geometric mean of $\sigma^2$ and $v^2$.

The learning rate $k_\infty$ of steady–state Kalman filter can also be simplified when the variance of observation noise $\sigma^2$ is larger than that for the process noise $v^2$. Using Eq 21, we find that

$$k_\infty \rightarrow \frac{v}{\sigma} \qquad\qquad 23$$

for $\frac{\sigma}{v} \rightarrow \infty$. This means that a steady–state Kalman filter with gain $\frac{v}{\sigma}$ is approximately optimal for signals with $\sigma^2 \gg v^2$. In Fig 5B, we compare the optimal steady–state learning rate $k_\infty$ with the approximately optimal learning rate $v/\sigma$ for different levels of $\sigma$, with $v$ fixed at $v = 1$. We find that the approximation becomes very close very quickly—for $\frac{\sigma}{v} > 2$, the relative difference between the optimal learning rate and its approximation is already less than 30%. Fig 5B further suggests that the approximation breaks down as $\frac{v}{\sigma}$ approaches unity—the optimal learning rate for signals with $\sigma = 0$ is one; any higher learning rate will be detrimental for the performance.

In summary, we find the learning rule

$$m_t = m_{t-1} + \frac{v}{\sigma}(r_t - m_{t-1}) \qquad\qquad 24$$

to be approximately optimal for $\sigma \ll v$ and large $t$. The rule Eq 24 bears striking resemblance to one of the SPE learning rules: the rule in Eq 3. The difference between the two rules is just how the scaling is attributed: in the Kalman filter, one would perhaps speak of a scaled learning rate, while in the SPE model, one attributes the scaling to the error term. Mathematically, both formulations are equivalent.

We conclude that the SPE model can be viewed as an implementation of approximately optimal one–dimensional state estimation, equipped with a mechanism to supply some of the required parameters—the observation noise $\sigma$.

## Reward prediction performance

In Fig 2, we compare the performance of the RW model, the SPE model and the Kalman filter. The SPE model was defined by the learning rules given in Eqs 2–4, the Kalman filter by the learning rules Eqs 17–19. The RW model was defined by

$$\delta = r - m_t \qquad\qquad 25$$

$$m_t = m_{t-1} + \alpha\delta \qquad\qquad 26$$

With a constant learning rate $\alpha$.

For Fig 2A, the RW model was parametrized with $\alpha = 0.5$ and the SPE learning rules were parametrized with $\alpha_m = 1$ and $\alpha_s = 0.1$. Rewards were sampled from a normal distribution

with drifting mean. The process noise was fixed at $v = 1$. We used three different observation noise levels (1, 5 and 15).

For Fig 2B, we used 10 different learning rates for the RW model (ranging from 0.007 to 0.993), the same parameters as above for the SPE model, and 100 different levels of observation noise, evenly distributed on a logarithmic scale from 0.1353 to 1096.6. The process noise was fixed at $v = 1$ as above. For each combination, we simulated $10^5$ trials and computed the average squared difference between the model predictions $m$ and the true mean $\mu$ across all trials. The Kalman filter model was parameterized with the true underlying process and observation noise parameters to ensure optimal performance. The posterior variance $w$ was initialized at $w_0 = 1$.

## Simulations of the task of Tobler et al

To simulate the relevant parts of the experiment reported by Tobler, Fiorillo [10], we modelled Pavlovian conditioning with three different stimuli, which were associated with three different reward magnitudes ($r = 0.05$, $r = 0.15$, $r = 0.5$). The stimuli were followed by the associated reward in one half of the trials and by no reward in the other half. The rewarded trials were selected pseudorandomly, such that there were two rewarded and two non–rewarded trials in every four successive trials.

We simulated 2000 trials per stimulus and extracted prediction errors from the last 1500. Discarding the first 500 trials accounts for the substantial pretraining of Tobler, Fiorillo [10].

We used two models: an RW model and a SPE model. The learning rules of the RW model are given in Eqs 25 and 26. The rules were used with $\alpha_m = 0.0067$ and $m_0 = 0$. The learning rules of the SPE model are given in Eqs 2–4. These rules were used with $\alpha_m = \alpha_s = 0.0067$ and $m_0 = 0$, $s_0 = 1$.

To compare our simulations to the experimental data from Tobler, Fiorillo [10], we extracted the prediction errors $\delta$ from the simulations and averaged them for each model, outcome and condition separately. There were three conditions (corresponding to the three reward sizes) with two outcomes (reward or no nothing) each, resulting in a total of six combinations per model. Finally, we normalized the six averaged prediction errors by their standard deviation for each model.

## A dynamical model of the basal ganglia

The differential equations Eqs 13 and 14 were solved using MATLAB's **ode15s**, from $t = -200$ $ms$ until $t = 500$ $ms$. As inputs, we used step functions

$$r(t) = \theta(t)r_{step} \qquad 27$$

$$G(t) = \theta(t)G_{step} \qquad 28$$

$$N(t) = \theta(t)N_{step} \qquad 29$$

with $\theta(t) = 1$ for $t>0$ and $\theta(t) = 0$ for $t<0$, and $G_{step} = 10$, $N_{step} = 6$, and $r_{step} = 4$. These inputs correspond to a learned mean $m = 2$ and a learned standard deviation $s = 8$ for $\lambda = 1$.

The time constant $\tau_T$ of the thalamic population was set to 10 ms, based on the measurement of the membrane time constant of thalamic neurons reported by Paz, Chavez [48]. The time constant $\tau_\delta$ for striatal dopamine was set to 300 ms, based on figure 2C of Montague, McClure [49]: the dopamine transient in that figure decays to exp−1 of its peak value in about 300 ms.

## Supporting information

**S1 Appendix. Derivation from Bayesian learning.** Fig A in S1 Appendix. The mode–matching method.
(DOCX)

## Author Contributions

**Investigation:** Moritz Möller.

**Writing – original draft:** Moritz Möller.

**Writing – review & editing:** Sanjay Manohar, Rafal Bogacz.

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
