## [Decision Letter · Decision Letter 0]

15 Feb 2022

Dear Dr. Bogacz,

Thank you very much for submitting your manuscript "Uncertainty-guided learning with scaled prediction errors in the basal ganglia" for consideration at PLOS Computational Biology.

As with all papers reviewed by the journal, your manuscript was reviewed by members of the editorial board and by several independent reviewers. In light of the reviews (below this email), we would like to invite the resubmission of a significantly-revised version that takes into account the reviewers' comments. The reviewers and I agree that this work is timely and interesting. There are two major issues to address, as detailed below: (1) A scholarship issue, regarding adequacy of the cited literature, and (2) a normative issue, regarding the link to Kalman filtering.

We cannot make any decision about publication until we have seen the revised manuscript and your response to the reviewers' comments. Your revised manuscript is also likely to be sent to reviewers for further evaluation.

Sincerely,

Samuel J. Gershman

Deputy Editor

PLOS Computational Biology

Reviewer's Responses to Questions

**Comments to the Authors:**

Reviewer #1: The authors propose in the present paper a neurocircuitry that might underlie the precision-weighting of prediction errors. The paper is well written, and their proposed model is interesting. Some sections constitute less of a novel contribution to the field though. Some rewriting and context would benefit particular sections, to clarify the novelty of the contributions.

The first part of the paper discusses their scale prediction error model, and perform simulations to demonstrate the model outperforms a more simple RW-model. This section is quite expansive, particularly since there have been many models that attempt to deal with uncertainty in prediction errors, and therefore does not constitute a novel contribution to the field. That is, researchers have covered the importance of precision-weighting of prediction errors, and have proposed various computational models that could account for such processes. For example see the work of Matthew Nassar, Chris Mathys, but also Schultz’s group, who worked on the Tobler findings that are central to this paper, who have done much work on precision-weighting of prediction errors in humans and the role of dopamine herein (For example see Diederen et al., 2015, 2016; 2017; Haarsma et al., 2020). I think the paper would benefit from mentioning this work when describing their particular model, to demonstrate what is novel about it, or why it might not be of importance to get into the differences between these models. I realise that later on in the discussion different models are discussed as well, but it’s not clear to me that their present model provides a clear advantage over other models that seem to capture the processes the authors are interested as well, like the hierarchical gaussian filter for example. If it’s the case that this section more acts as a proof of principle that precision-weighting is important, rather than coming up with a novel insight in reinforcement learning, I would suggest shortening this section and emphasise this point.

The second section discusses how the striatum might compute precision-weighted prediction errors by largely expanding on previous work by the same group on learning reward uncertainty. I think the model is of interest and constitutes a novel contribution to the field on how these mechanisms might be implicated in striatal sub-cortical loops. Some more discussion on how their model compares to other approaches might be beneficial here as well.

Reviewer #2: The authors present a new model for learning in noisy environments. The work is mostly focused on presenting a model which is biologically plausible and authors also present a detailed neuronal mechanism of the model in the basal ganglia. I think this is a timely question and the proposed solution is interesting and novel. The paper is well-written. I’m overall positive about this paper with one major caveat that I hope authors can sufficiently address.

The main issue with the paper is about its link to Kalman filtering, which is important within the paper because authors present their model as a variant of Kalman filter. Authors argue that whereas the line of Kalman filtering work in the literature posits that organisms should be able to track observation noise, there is no biologically-realistic computational model for that. But the main idea behind Kalman filtering is optimal tracking, which requires tracking both posterior mean and posterior uncertainty. The proposed model does not track posterior uncertainty. Authors seem to ignore posterior uncertainty altogether, which is technically the most important facet of Kalman filtering (when both process- and observation noise are fixed but are nonzero.) In fact, I believe (but I’m happy to be convinced otherwise) that their model actually recovers the posterior uncertainty but authors mistakenly label it as observation noise. That’s because they assume that zero process noise in their generative process (fixed mu in Eq 1). In Appendix S2, authors show that the model is equivalent to the Kalman filter when the ratio of observation- to process- noise is infinite. That only happens if one assumes that the process noise is zero. These are all fine for a descriptive model, but I think these assumptions ruin the link to the normative learning mechanism of Kalman filter. And the main problem is that it is not clear why we do need to make such a huge sacrifice from the normative point: does tracking posterior uncertainty really require some process that are so different from the current model?

Perhaps the strongest aspect of the proposed model is the basal ganglia mechanism for implementing the model at the neuronal level. I found that part timely and novel, but again I believe the same neuronal mechanism can be told for a model that scales prediction error with posterior uncertainty in the more general case that the process noise is nonzero.

A number of minor points:

- I suggest to move the proof in the “fixed point analysis” section to Methods to enhance readability of the paper for broader audience. I think that part is quite trivial for technical readers, and quite confusing for general psychologists/neuroscientists; so in both case does not help to present it in the forefront of results section.

- I found the discussion on the free energy models too vague. Maybe authors can explain more about those models and their relation to this work, especially given the recent work by Bogacz (2020).

- I also found the discussion on AI models quite speculative. I think there might be a weak connection between these models and deep RL models, but that is very non-specific (i.e. many other biological learning model might have similar connections). I think deep RL models are not competitors of the current model. Also they do not help to understand the current work.

- In the introduction, authors seem to imply that particle filtering is not biologically realistic without providing any reason why that is so (quite in contrary, there are influential papers arguing plausibility of Monte Carlo models for biological learning, eg Courville and Daw 2006; Gershamn et al, 2010). Relatedly, it is stated in the introduction that the model by ref 9 is not suitable to describe biological learning. While it is true that the focus of that work is the computational level, but that does not mean that it is not suitable to describe biological learning on the mechanistic or algorithmic level, especially given that it is a combination of particle filtering and Kalman filtering.

**Have the authors made all data and (if applicable) computational code underlying the findings in their manuscript fully available?**

Reviewer #1: Yes

Reviewer #2: None

PLOS authors have the option to publish the peer review history of their article (what does this mean?). If published, this will include your full peer review and any attached files.

Reviewer #1: No

Reviewer #2: No
---

## [Decision Letter · Decision Letter 1]

5 May 2022

Dear Dr. Bogacz,

We are pleased to inform you that your manuscript 'Uncertainty-guided learning with scaled prediction errors in the basal ganglia' has been provisionally accepted for publication in PLOS Computational Biology.

Best regards,

Samuel J. Gershman

Deputy Editor

PLOS Computational Biology

Reviewer's Responses to Questions

**Comments to the Authors:**

Reviewer #2: Authors have revised the manuscript sufficiently and have addressed my comments.

**Have the authors made all data and (if applicable) computational code underlying the findings in their manuscript fully available?**

Reviewer #2: None

PLOS authors have the option to publish the peer review history of their article (what does this mean?). If published, this will include your full peer review and any attached files.

Reviewer #2: No

---

## [Editor Report · Acceptance letter]

23 May 2022

PCOMPBIOL-D-22-00036R1 

Uncertainty-guided learning with scaled prediction errors in the basal ganglia

Dear Dr Bogacz,

I am pleased to inform you that your manuscript has been formally accepted for publication in PLOS Computational Biology. Your manuscript is now with our production department and you will be notified of the publication date in due course.

With kind regards,

Livia Horvath
